# SELFI: Selective Fusion of Identity for Generalizable Deepfake Detection

## Abstract

Face identity provides a remarkably powerful signal for deepfake detection. Prior studies have shown that even when not explicitly modeled, deepfake classifiers tend to implicitly learn identity features during training. This has led to two conflicting viewpoints in the literature: some works attempt to completely suppress identity cues to mitigate bias, while others rely on them exclusively as a strong forensic signal. To reconcile these opposing stances, we conduct a detailed empirical analysis based on two central hypotheses: (1) whether face identity alone is inherently discriminative for detecting deepfakes, and (2) whether such identity features generalize poorly across manipulation methods. Through extensive experimentation, we confirm that face identity is indeed a highly informative signal—but its utility is context-dependent. While some manipulation methods preserve identity-consistent artifacts, others distort identity cues in ways that can harm generalization. These findings suggest that identity features should not be suppressed or relied upon blindly. Instead, they should be explicitly modeled and adaptively controlled based on their per-sample relevance. To this end, we propose **SELFI** (**SEL**ective **F**usion of **I**dentity), a generalizable deepfake detection framework that dynamically modulates identity usage. SELFI consists of: (1) a Forgery-Aware Identity Adapter (FAIA) that explicitly extracts face identity embeddings from a frozen face recognition model and projects them into a forgery-relevant space using auxiliary supervision, and (2) an Identity-Aware Fusion Module (IAFM) that selectively integrates identity and visual features via a relevance-guided fusion mechanism. Extensive experiments on four benchmark datasets demonstrate that SELFI achieves strong generalization across manipulation methods and datasets, outperforming prior state-of-the-art methods by an average of 3.1% frame-level AUC in cross-dataset evaluations. Notably, on the challenging DFDC benchmark, SELFI improves over the previous best by a significant 6% margin, highlighting the effectiveness of adaptive identity control. The code will be released upon acceptance of the paper.

## 1 Introduction

Recent advances in deepfake generation Kowalski (2018); Thies et al. (2016); Li et al. (2019); FaceSwapDevs (2019) have raised serious concerns about the authenticity of visual media, driving a surge in research on robust and generalizable detection methods Yan et al. (2024; 2023a); Dong et al. (2022); Cheng et al. (2024b;a). While many existing detectors perform well on seen manipulation types, they often struggle to generalize to unseen forgeries—a critical limitation for real-world deployment Afchar et al. (2018); Li & Lyu (2018); Yang et al. (2019); Qian et al. (2020). One possible explanation for this generalization gap lies in the treatment of face identity during training. Dong et al. Dong et al. (2023) attribute poor generalization to implicit bias caused by face identity: because identity is such a strong discriminative signal, models tend to rely on it unconsciously, leading to overfitting to identity-specific patterns in the training data. In contrast, Huang et al. Huang et al. (2023) argue that face identity itself is a powerful forensic cue, and that explicitly leveraging identity embeddings during training enhances robustness. These two opposing views—treating identity as a harmful bias versus a beneficial signal—highlight the need for a deeper understanding of how face identity influences detection performance across manipulation types.

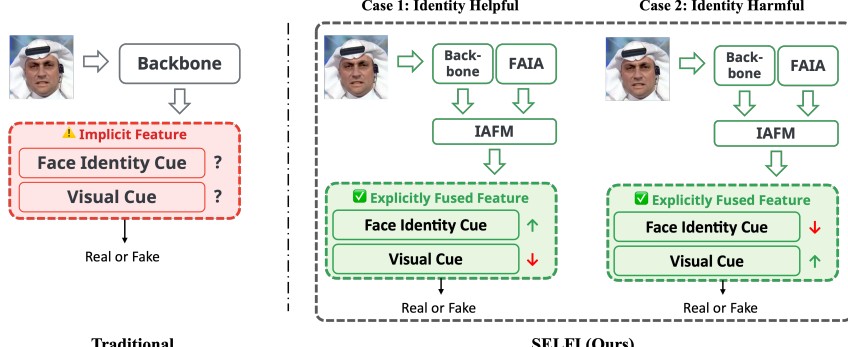

Figure 1: **Overview of the proposed SELFI framework.** Traditional deepfake detectors implicitly rely on identity cues without control, risking overfitting and poor generalization across manipulation types. In contrast, SELFI explicitly incorporates face identity features through a Forgery-Aware Identity Adapter (FAIA) and adaptively fuses them with visual features using an Identity-Aware Fusion Module (IAFM). This enables the model to leverage identity information when helpful and suppress it when harmful, resulting in more robust and generalizable deepfake detection.

To address this, we conduct an in-depth analysis guided by two core hypotheses. First, we examine whether face identity alone can meaningfully support deepfake detection. Specifically, we extract identity embeddings from a pretrained face recognition model (IResNet100 Deng et al. (2019)) that has never seen manipulated data, and train a lightweight classifier on top. Despite the absence of visual content and deepfake-specific supervision, we observe strong classification performance, confirming that face identity is inherently discriminative for this task. Second, we investigate whether such identity features generalize across manipulation methods, or if they encode method-specific patterns that hinder transferability. Using the FaceForensics++ Rossler et al. (2019) dataset, we perform a cross-manipulation experiment where models trained on one type of forgery are tested on the others. This reveals three distinct identity behavior categories: (1) transferable identity cues, observed in manipulations like DeepFakes FaceSwapDevs (2019) and FaceSwap Kowalski (2018), where identity information remains relatively intact and generalizes well; (2) method-specific identity artifacts, found in Face2Face Thies et al. (2016), where identity features are entangled with generation-specific artifacts and do not transfer; and (3) ineffective identity cues, as in NeuralTextures Thies et al. (2019), where identity features remain ambiguous, offering limited discriminative value for detection. These observations collectively indicate that the utility of identity features is highly context-dependent, and thus demand a more flexible, selective integration mechanism. In particular, while face identity is undeniably a powerful signal that should not be disregarded, allowing the model to implicitly absorb it without control can lead to generalization issues, especially when identity cues interact differently across manipulation methods.

These observations collectively indicate that the utility of identity features is highly context-dependent, and thus demand a more flexible, selective integration mechanism. In particular, while face identity is undeniably a powerful signal that should not be disregarded, allowing the model to implicitly absorb it without control can lead to generalization issues, especially when identity cues interact differently across manipulation methods. To explicitly address the challenges highlighted by this analysis regarding the context-dependent nature of identity cues and the necessity for selective integration, we design a solution that enables explicit and context-aware identity control. We propose SELFI (**SEL**ective **F**usion of **I**dentity), a novel framework that dynamically combines face identity features and backbone features to produce highly discriminative yet generalizable representations by adaptively leveraging identity based on its estimated relevance. SELFI consists of two core modules: (1) the **F**orgery-**A**ware **I**dentity **A**dapter (FAIA), which projects identity embeddings extracted from a frozen face recognition model into a forgery-relevant representation space using a learnable transformation matrix. This projection is guided by an auxiliary supervision signal, the *Forgery-Aware Guidance Loss*, which encourages the projected features to be discriminative for real-vs-fake classification even without visual cues, thereby making the identity information more pertinent to the detection task. (2) the **I**dentity-**A**ware **F**usion **M**odule (IAFM), which adaptively integrates identity and visual features. IAFM includes a *Relevance Predictor*, a lightweight neural network that estimates the importance of identity cues for each input, and a *Soft Fusion Operator*

that combines the two feature types via a weighted sum based on the predicted relevance score. This design allows SELFI to amplify identity information when it provides genuine forensic signals, and to downweight it when it risks introducing manipulation-specific bias, directly implementing the selective integration mechanism identified as necessary. The resulting fused feature representation jointly captures content and identity cues in a task-adaptive manner, enabling robust and generalizable deepfake detection. Fig. 1 illustrates the overall comparison between traditional approaches and our SELFI framework under different identity relevance scenarios.

- We conduct a quantitative analysis using a pretrained face recognition model, showing that identity features alone can distinguish real from fake content, and further identify three behavioral patterns across manipulation types: transferable, method-specific identity cues, and ineffective.

- Motivated by this observation, we propose SELFI, a framework that explicitly separates identity features from visual content and adaptively fuses them through two modules: FAIA, which projects identity embeddings into a forgery-discriminative space, and IAFM, which dynamically adjusts their contribution based on per-sample relevance.

- Extensive experiments demonstrate that SELFI outperforms existing methods in both in-domain and cross-domain settings. Specifically, SELFI achieves an average improvement of 3.1% in frame-level AUC across four cross-dataset benchmarks (CDFv2, DFD, DFDC, DFDCP), and surpasses the previous best on DFDC by a significant 6.0% margin.

## 2 RELATED WORKS

**Deepfake Detection and Generalization.** Early deepfake detection approaches primarily target visual artifacts introduced by manipulation techniques, such as blending boundaries Li et al. (2020a), inconsistent textures Wang & Chow (2023), and abnormal frequency patterns Gu et al. (2022a); Liu et al. (2021); Luo et al. (2021); Wang et al. (2023a). While these methods achieve promising performance on seen manipulation types, they often struggle to generalize to unseen forgeries or datasets due to their reliance on spurious and non-transferable cues. To enhance generalization, various strategies emerge. Frequency-domain methods aim to capture spectral inconsistencies between real and fake content Qian et al. (2020); Durall et al. (2020); Luo et al. (2021); Wang et al. (2023a); Li et al. (2024), while spatiotemporal modeling approaches leverage temporal dynamics and cross-modal signals to detect subtle manipulation traces Wang et al. (2023b); Xu et al. (2023); Gu et al. (2022b). Other studies employ latent space augmentation Yan et al. (2024); Choi et al. (2024) or self-blending techniques Shiohara & Yamasaki (2022); Li et al. (2020a) to diversify training data, and disentangle task-relevant features using representation separation Yan et al. (2023a); Yang et al. (2023) or progressive regularization Cheng et al. (2024b). Nevertheless, most of these methods focus predominantly on low-level visual cues, overlooking face identity—a semantically rich and manipulation-sensitive signal that remains underexplored despite its potential to offer deeper insights into forgery characteristics.

**Face Identity in Deepfake Detection.** Several studies explore the role of facial identity in deepfake detection, revealing two contrasting perspectives. One line of research actively leverages face identity as a strong semantic cue to improve detection robustness. For instance, Dong et al. Dong et al. (2022) propose ICT, an identity consistency transformer that measures coherence between inner and outer face regions, while Huang et al. Huang et al. (2023) introduce IID, which defines implicit identity to detect face swapping by contrasting it with explicit identity embeddings. These methods demonstrate that explicitly modeling identity cues reveals forgery-specific inconsistencies. However, overreliance on face identity may cause models to overlook important visual cues, limiting their ability to capture manipulation artifacts beyond identity mismatches. In contrast, other works identify facial identity as a source of bias that harms generalization across datasets. Dong et al. Dong et al. (2023) show in IIL that binary classifiers unintentionally learn identity boundaries, leading to implicit identity leakage and overfitting. Similarly, Kim et al. Kim et al. (2024) propose FRIDAY, which suppresses facial identity embeddings during training by encouraging the detector to diverge from a frozen face recognizer, thereby improving cross-domain robustness.

Motivated by these opposing findings, we aim to reconcile the contrasting views on face identity by conducting a systematic analysis grounded in two key hypotheses: (1) whether identity embeddings alone carry sufficient forensic information to detect manipulations, and (2) whether such features

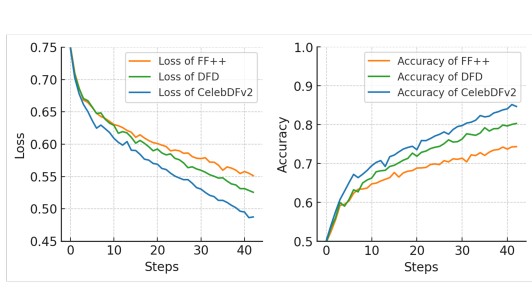

(a) Training Loss and Accuracy

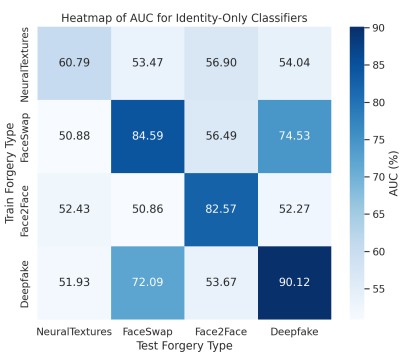

(b) Cross-Manipulation AUC

Figure 2: **Preliminary Analysis of Identity-Only Detection.** (a) Training loss and accuracy curves for identity-only detection using a frozen face recognition model (IResNet100) on FF++, DFD, and CelebDFv2. The results show consistent decreases in loss and increases in accuracy, indicating that face identity alone provides strong discriminative power for real-vs-fake classification. (b) Cross-manipulation AUC heatmap, where rows indicate training forgery types and columns indicate test types. The results reveal that identity features are transferable for some manipulations (DF and FS), but highly method-specific (F2F) or ineffective for others (NT), highlighting the need for explicit identity control.

generalize reliably across different forgery methods. Our empirical findings reveal that face identity is indeed a highly discriminative signal, but its utility varies significantly depending on the manipulation context.

## 3 PRELIMINARY ANALYSIS

### 3.1 THE POWER OF FACE IDENTITY IN DEEPFAKE DETECTION

**Hypothesis 1.** Face identity itself serves as a highly discriminative signal for deepfake detection, even without explicit modeling of forgery artifacts.

While face identity has long been considered an important cue for deepfake detection—albeit often implicitly—recent studies have expressed conflicting views on its role Dong et al. (2023); Huang et al. (2023); Kim et al. (2024); Dong et al. (2022). Some suggest that identity is a reliable signal for detecting manipulations, while others argue that reliance on identity introduces undesirable bias that hinders generalization. Motivated by this controversy, we aim to explicitly evaluate whether face identity alone carries sufficient forensic information to support deepfake detection.

To this end, we formulate Hypothesis 1 and conduct an experiment using identity embeddings extracted from a pretrained face recognition model that has never been exposed to deepfake data. Specifically, we adopt IResNet100 Deng et al. (2019), a widely used and well-established identity representation model, and freeze all of its parameters. We then train only a lightweight classifier head on top of the identity embeddings to assess whether face identity alone can support deepfake classification. Detailed experimental settings, architectural illustration, and analysis of the training outcomes beyond loss or accuracy curves are provided in Sec. A.2.

We conduct training and evaluation on three widely-used deepfake detection benchmarks: Face-Forensics++ Rossler et al. (2019), Celeb-DF v2 Li et al. (2020b), and DeepFakeDetection AI (2019). As shown in Fig. 2 (a), the classification loss consistently decreases and the accuracy steadily increases across all datasets, even with the backbone completely frozen. These result provide strong evidence that face identity contains inherently discriminative signals for real-vs-fake classification, and can serve as a standalone foundation for deepfake detection models. We next investigate whether the identity features captured in this setting remain robust across different manipulation types, or if they instead encode method-specific biases that compromise generalization.

## 3.2 The Generalization Risk of Identity-Specific Features

**Hypothesis 2.** Different deepfake generation methods manipulate face identity in distinct ways. Therefore, over-reliance on identity cues that work well for specific methods may hinder generalization to unseen manipulations.

To test this hypothesis, we conduct a cross-manipulation experiment using the FaceForensics++ dataset Rossler et al. (2019), which includes four representative manipulation types: Deep-Fakes FaceSwapDevs (2019), FaceSwap Kowalski (2018), Face2Face Thies et al. (2016), and NeuralTextures Thies et al. (2019). For each method, we train a classifier using identity embeddings extracted from a frozen IResNet100 model, and evaluate both in-domain and cross-domain performance. For example, a model trained on DeepFakes is tested not only on DeepFakes but also on the other three types.

The results, illustrated in Fig. 2 (b), reveal three distinct identity behavior patterns: (1) *Transferable identity cues*, as seen in DeepFakes and FaceSwap, where identity information is well preserved and generalizes effectively across methods (e.g., $84.6\% \rightarrow 74.5\%$); (2) *Method-specific identity artifacts*, as in Face2Face, where in-domain performance is high ($82.6\%$) but generalization is poor (e.g., $52.3\%$ on DeepFakes), indicating overfitting to generation-specific patterns; and (3) *Ineffective identity cues*, as in NeuralTextures, which produces weak or distorted identity features, leading to both low in-domain ($60.8\%$) and cross-domain (e.g., $53.5\%$) performance.

These findings demonstrate that the effectiveness of identity features is highly dependent on the manipulation method. Some techniques preserve identity in a way that supports generalization, while others entangle identity with artifacts or suppress it entirely. Consequently, uniformly relying on identity cues can harm generalization when such cues encode method-specific biases.

We argue that a more robust strategy is to explicitly separate identity from visual content and adaptively regulate its use based on context. This insight directly motivates the design of our proposed framework, **SELFI**, which selectively integrates identity features according to their per-sample relevance.

## 4 Proposed Method

Motivated by the observations in Sec. 3, we introduce **SELFI** (SELective Fusion of Identity), a framework designed for generalizable deepfake detection by adaptively fusing identity and visual cues. As illustrated in Fig. 3, SELFI incorporates a **Forgery-Aware Identity Adapter (FAIA)** to transform identity embeddings into a forgery-relevant space and an **Identity-Aware Fusion Module (IAFM)** to dynamically integrate these with visual features based on estimated relevance. This adaptive fusion strategy allows SELFI to leverage reliable identity signals while suppressing biases, thereby enhancing generalization across diverse manipulation types, with detailed module designs presented in the following subsections.

### 4.1 Forgery-Aware Identity Adapter (FAIA)

The goal of FAIA is to extract face identity information from an input image and transform it into a representation that is more suitable for deepfake detection. While identity features from pretrained face recognition models are optimized for verifying who appears in an image, our preliminary analysis reveals that they can already serve as a surprisingly strong signal for deepfake detection. However, these embeddings are not specifically tailored for capturing forgery-related artifacts. To better align identity features with manipulation cues, FAIA learns to project them into a forgery-aware space where their utility for classification is maximized.

FAIA operates in two main stages. First, we extract a face identity embedding from the input image using a frozen face recognition model $\mathcal{E}_{id}$:

$$\mathbf{f}_{id} = \mathcal{E}_{id}(\mathbf{x}), \tag{1}$$

where $\mathbf{x}$ is the input image and $\mathbf{f}_{id} \in \mathbb{R}^{D_{id}}$ is the identity embedding. Here, $D_{id}$ denotes the output dimensionality of the face recognition model. In our study, we use IResNet100 Deng et al. (2019), for which $D_{id} = 512$.

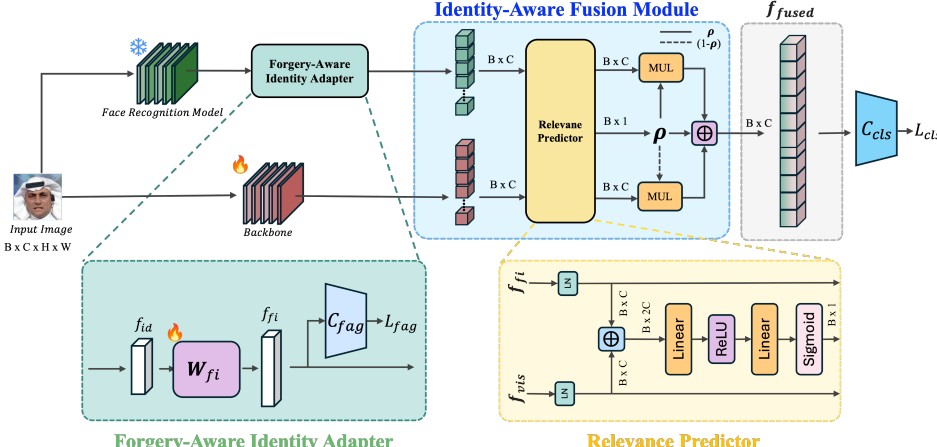

Figure 3: **Architecture of the proposed SELFI framework.** SELFI consists of two main modules: (1) the Forgery-Aware Identity Adapter (FAIA), which projects frozen identity embeddings into a forgery-relevant space using a trainable transformation and auxiliary supervision (Sec. 4.1); and (2) the Identity-Aware Fusion Module (IAFM), which adaptively fuses identity and visual features based on a predicted relevance score (Sec. 4.2). By explicitly modeling and controlling identity usage, SELFI enables robust and generalizable deepfake detection across diverse manipulation types.

Next, we project this identity embedding into the backbone feature space using a trainable weight matrix $W_{\text{fi}} \in \mathbb{R}^{D_{\text{backbone}} \times D_{\text{id}}}$, where $D_{\text{backbone}}$ denotes the feature dimensionality of the visual backbone. In our implementation, we use CLIP Radford et al. (2021) as the backbone, for which $D_{\text{backbone}} = 768$ (see Sec. A.8 for the rationale behind choosing trainable weight matrix instead of MLP):

$$\mathbf{f}_{\text{fi}} = W_{\text{fi}}\mathbf{f}_{\text{id}}. \tag{2}$$

To guide the transformation $W_{\text{fi}}$ to produce identity features that are discriminative for forgery detection, we introduce an auxiliary supervision called the *Forgery-Aware Guidance Loss*. Specifically, we attach a lightweight binary classifier $\mathcal{C}_{\text{fag}}$ to the projected feature $\mathbf{f}_{\text{fi}}$ and train it using the standard CrossEntropy loss:

$$\mathcal{L}_{\text{fag}} = \text{CE}(\mathcal{C}_{\text{fag}}(\mathbf{f}_{\text{fi}}), y), \tag{3}$$

where $y \in \{0, 1\}$ indicates whether the input is real or fake.

This auxiliary supervision encourages $W_{\text{fi}}$ to produce forgery-aware identity embeddings that are semantically aligned with manipulation cues. The resulting projected feature $\mathbf{f}_{\text{fi}} \in \mathbb{R}^{D_{\text{backbone}}}$ is then fused with visual features in the Identity-Aware Fusion Module (IAFM).

## 4.2 IDENTITY-AWARE FUSION MODULE (IAFM)

While forgery-aware identity features can be informative, they are not equally useful across all samples or manipulation types. Over-reliance on identity cues in irrelevant contexts may even hurt generalization. To address this, IAFM adaptively integrates identity and visual features by estimating the relevance of identity information for each input.

Given the projected identity feature $\mathbf{f}_{\text{fi}} \in \mathbb{R}^{D_{\text{backbone}}}$ from FAIA and the visual feature $\mathbf{f}_{\text{vis}} \in \mathbb{R}^{D_{\text{backbone}}}$ extracted from the visual encoder (CLIP Radford et al. (2021)), we first concatenate the two and feed them into a lightweight relevance predictor (see Sec. A.7 for the rationale behind choosing concatenation):

$$\rho = \mathcal{R}([\mathbf{f}_{\text{vis}}; \mathbf{f}_{\text{fi}}]), \tag{4}$$

where $\mathcal{R}(\cdot)$ denotes a small feedforward network that outputs a scalar relevance score $\rho \in [0, 1]$ via a sigmoid activation. In our implementation, $\mathcal{R}(\cdot)$ consists of two linear layers with ReLU activation in between, followed by a sigmoid at the end:

$$\mathcal{R}(\cdot) = \text{Sigmoid}(\text{Linear}_2(\text{ReLU}(\text{Linear}_1(\cdot)))).$$

Table 1: **Frame-level AUC Performance.** All models are trained on FF++ c23 Rossler et al. (2019). The best and second-best results are highlighted in **bold** and underlined, respectively. Reported scores for prior methods are taken from DeepfakeBench Yan et al. (2023b) and their original publications Cheng et al. (2024b); Yan et al. (2024); Kashiani et al. (2025). A dash (–) indicates that the corresponding result is not available in the original papers.

| Method | Publication | FF++ | CDFv2 | DFD | DFDC | DFDCP | C-Avg. |
|---|---|---|---|---|---|---|---|
| Xception Chollet (2017) | CVPR'17 | 0.964 | 0.737 | 0.816 | 0.708 | 0.737 | 0.750 |
| Meso4 Afchar et al. (2018) | WIFS'18 | 0.608 | 0.609 | 0.548 | 0.556 | 0.599 | 0.578 |
| FWA Li & Lyu (2018) | CVPRW'18 | 0.877 | 0.668 | 0.740 | 0.613 | 0.638 | 0.665 |
| EfficientB4 Tan & Le (2019) | ICML'19 | 0.957 | 0.749 | 0.815 | 0.696 | 0.728 | 0.747 |
| Capsule Nguyen et al. (2019) | ICASSP'19 | 0.842 | 0.747 | 0.684 | 0.647 | 0.657 | 0.684 |
| X-ray Li et al. (2020a) | CVPR'20 | 0.959 | 0.679 | 0.766 | 0.633 | 0.694 | 0.693 |
| F3Net Qian et al. (2020) | ECCV'20 | 0.964 | 0.735 | 0.798 | 0.702 | 0.765 | 0.750 |
| SPSL Liu et al. (2021) | CVPR'21 | 0.961 | 0.765 | 0.812 | 0.704 | 0.741 | 0.756 |
| SRM Luo et al. (2021) | CVPR'21 | 0.958 | 0.755 | 0.812 | 0.700 | 0.741 | 0.752 |
| CORE Ni et al. (2022) | CVPRW'22 | 0.964 | 0.743 | 0.802 | 0.705 | 0.734 | 0.746 |
| Recce Cao et al. (2022) | CVPR'22 | 0.962 | 0.732 | 0.812 | 0.713 | 0.742 | 0.750 |
| UCF Yan et al. (2023a) | ICCV'23 | 0.971 | 0.753 | 0.807 | 0.719 | 0.759 | 0.760 |
| OPR Cheng et al. (2024b) | NeruIPS'24 | 0.959 | **0.845** | - | 0.724 | 0.812 | - |
| LSDA Yan et al. (2024) | CVPR'24 | - | 0.830 | 0.880 | 0.736 | 0.815 | 0.815 |
| FreqDebias Kashiani et al. (2025) | CVPR'25 | 0.975 | 0.836 | 0.868 | 0.741 | 0.824 | 0.817 |
| SELFI (Ours) | | **0.980** | 0.839 | **0.907** | **0.796** | **0.840** | **0.846** |

We then compute the final fused representation as a weighted combination of the identity and visual features:

$$\mathbf{f}_{\text{fused}} = \rho \cdot \mathbf{f}_{\text{fi}} + (1 - \rho) \cdot \mathbf{f}_{\text{vis}}. \tag{5}$$

This soft fusion strategy enables the model to conditionally emphasize identity or visual cues depending on their relevance to the input, improving both flexibility and generalization. Finally, the fused feature $\mathbf{f}_{\text{fused}}$ is passed to the final classification head $\mathcal{C}_{\text{cls}}$ to produce the prediction:

$$\hat{y} = \mathcal{C}_{\text{cls}}(\mathbf{f}_{\text{fused}}), \tag{6}$$

where $\hat{y} \in [0, 1]$ represents the probability of the input being a fake.

### 4.3 OVERALL LOSS FUNCTION

We jointly optimize two objectives: the main classification loss based on the fused representation and an auxiliary guidance loss applied to the projected identity features. The overall loss is defined as:

$$\mathcal{L}_{\text{total}} = \alpha \cdot \mathcal{L}_{\text{cls}} + \beta \cdot \mathcal{L}_{\text{fag}}, \tag{7}$$

where $\mathcal{L}_{\text{cls}}$ is the standard CrossEntropy loss on the final prediction from $\mathcal{C}_{\text{cls}}$, and $\mathcal{L}_{\text{fag}}$ supervises the identity projection via $\mathcal{C}_{\text{fag}}$. We use $\alpha = 1.0$ and $\beta = 1.0$ in all experiments.

## 5 EXPERIMENTS

### 5.1 EXPERIMENTAL SETTING

**Datasets.** We train our models on the FaceForensics++ (FF++) dataset Rossler et al. (2019), using the c23 (lightly compressed) version comprising four manipulation types: DeepFakes (DF) FaceSwapDevs (2019), Face2Face (F2F) Thies et al. (2016), FaceSwap (FS) Kowalski (2018), and NeuralTextures (NT). To evaluate cross-dataset generalization, we test on four disjoint datasets: Celeb-DF v2 (CDFv2) Li et al. (2020b), DeepfakeDetection (DFD) AI (2019), the Deepfake Detection Challenge (DFDC) and its preview version (DFDCP) Kaggle. These benchmarks differ in content, manipulation methods, and post-processing, providing a comprehensive generalization testbed.

### 5.2 OVERALL PERFORMANCE ON COMPREHENSIVE DATASETS

We follow the DeepfakeBench Yan et al. (2023b) framework, with detailed settings provided in Appendix A.1.

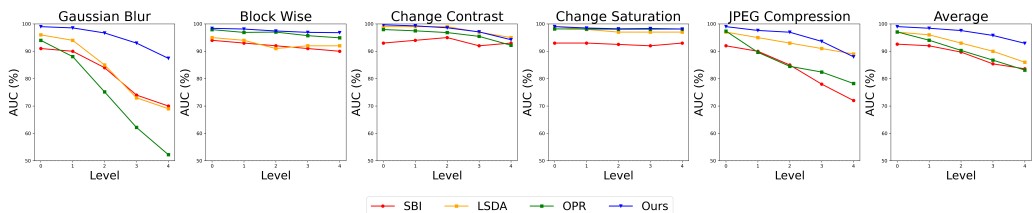

Figure 4: **Robustness to Unseen Perturbations.** We report video-level AUC (%) across five degradation levels for five types of perturbations Jiang et al. (2020).

Table 2: **Ablation study of SELFI modules with CLIP backbone.** FAIA-only denotes simple feature concatenation without adaptive fusion. (see Sec. A.4 for details)

| Modules | | | CLIP | | | |
|---|---|---|---|---|---|---|
| FAIA | $\mathcal{L}_{\text{fag}}$ | IAFM | CDFv2 | DFD | DFDCP | Avg |
| ✗ | ✗ | ✗ | 0.795 | 0.830 | 0.793 | 0.806 |
| ✓ | ✗ | ✗ | 0.804 | 0.887 | **0.844** | 0.845 |
| ✓ | ✗ | ✓ | 0.819 | 0.899 | 0.830 | 0.849 |
| ✓ | ✓ | ✓ | **0.839** | **0.907** | 0.840 | **0.862** |

Table 3: **Cross-dataset AUC of SELFI with different backbones.** SELFI consistently improves generalization across all architectures.

| Backbones | CDFv2 | DFD | DFDCP | Avg |
|---|---|---|---|---|
| CLIP | 0.795 | 0.830 | 0.793 | 0.806 |
| +SELFI | **0.839** | **0.907** | **0.840** | **0.862** |
| ResNet34 | 0.739 | 0.810 | 0.690 | 0.746 |
| +SELFI | **0.762** | **0.822** | **0.731** | **0.772** |
| EfficientB4 | 0.747 | **0.823** | 0.685 | 0.751 |
| +SELFI | **0.755** | 0.812 | **0.733** | **0.767** |

**Frame-level Generalization Performance.** As shown in Tab. 1, we evaluate our model by training on FF++ and testing both in-domain (FF++) and cross-domain (CDFv2, DFD, DFDC, DFDCP) datasets. Our method achieves superior performance in the in-domain setting compared to all existing approaches. In the more challenging cross-domain scenario, our model outperforms all prior methods on every dataset except for CDFv2, where it ranks second. Notably, compared to LSDA Yan et al. (2024)—one of the most recent and competitive state-of-the-art detectors—our approach achieves an average improvement of 3.5% frame-level AUC across the cross-domain benchmarks, highlighting its strong generalization capability. A comparison with video-level deepfake detectors is provided in Sec. A.3.

**Robustness to Unseen Perturbations.** To evaluate the robustness of our method against unseen perturbations, we conducted experiments using five types of perturbations introduced in DeeperForensics Jiang et al. (2020) and compared our approach with three recent state-of-the-art deepfake detection models: OPR Cheng et al. (2024b), LSDA Yan et al. (2024), and SBI Shiohara & Yamasaki (2022). As shown in Figure 4, our method consistently achieves competitive or superior video-level AUC scores across all perturbation types and degradation levels, showing a particularly large margin under Gaussian Blur, which severely degrades high-frequency forgery cues. Moreover, when averaged across all perturbation types and levels, our method outperforms existing models, demonstrating its stronger generalization ability to previously unseen corruptions. While baseline models exhibit substantial performance degradation as the perturbation level increases, our method maintains relatively stable performance, highlighting its robustness to distribution shifts caused by unseen perturbations.

## 5.3 Ablation Study

**Module-Wise Contribution Analysis.** To verify the contribution of each component within SELFI, we perform a step-by-step ablation using CLIP as the backbone (Tab. 2). Incorporating only the Forgery-Aware Identity Adapter (FAIA), where identity features are projected and simply concatenated with backbone features before classification, improves average AUC by 3.9% over the baseline. This suggests that even unmodulated identity features can serve as useful cues when aligned with visual content. The detailed architecture of this configuration is described in Sec. A.4. Introducing the Identity-Aware Fusion Module (IAFM), which adaptively weighs the relevance of identity information, further improves performance by 0.4%, highlighting the benefit of conditional identity integration. Finally, when supervised with the forgery-specific guidance loss $\mathcal{L}_{\text{fs}}$, the full SELFI framework achieves an additional 1.3% gain, validating the synergistic effect of jointly learning forgery-aware identity representations and adaptive fusion. For additional ablation results on other backbones, please refer to Sec. A.4.

Table 4: **Effect of auxiliary feature sources on frame-level AUC.** Only identity-aware features improve generalization, while others offer limited benefit.

| Auxiliary Source | Backbone | FF++ | CDFv2 | DFD | DFDC | DFDCP | C-Avg. |
|---|---|---|---|---|---|---|---|
| X (No Fusion) | - | 0.979 | 0.795 | 0.830 | **0.810** | 0.793 | 0.807 |
| Random Intialization | ResNet101 | 0.542 | 0.544 | 0.513 | 0.552 | 0.581 | 0.548 |
| ImageNet | ResNet101 | 0.726 | 0.586 | 0.650 | 0.617 | 0.615 | 0.617 |
| Deepfake-Trained | OPR Kashiani et al. (2025) | 0.975 | **0.856** | 0.874 | 0.767 | 0.795 | 0.823 |
| Face Identity | IResNet100 Deng et al. (2019) | **0.980** | 0.839 | **0.907** | 0.796 | **0.840** | **0.846** |

**Effect of Auxiliary Feature Source.** To verify that the performance gain of SELFI does not stem from a simple ensemble effect but from the use of face identity features, we designed an ablation study with four different auxiliary sources: (1) *Random Initialization*, representing purely uninformative features; (2) *ImageNet*, trained on generic object recognition tasks and not specific to faces; (3) *Deepfake-Trained*, trained for forgery detection but lacking explicit identity information (we adopted features from OPR Kashiani et al. (2025)); and (4) *Face Identity*, our proposed identity feature extracted from a frozen face recognition model. We replaced only the frozen face recognition model in the SELFI architecture while keeping the rest of the framework fixed. As shown in Tab. 10, identity-aware features lead to the best generalization, boosting the cross-dataset average AUC by 23.8% over Random, 22.9% over ImageNet, and 2.3% over Deepfake-Trained features. These findings confirm that the fusion benefits of SELFI arise not from the ensemble size but from the quality and relevance of the auxiliary signal. This highlights that identity semantics, rather than mere forgery correlations, are essential for achieving robust and transferable deepfake detection.

**Generalization Across Backbones.** To evaluate the versatility of SELFI, we integrate it with three different backbone architectures—CLIP, ResNet34, and EfficientNet-B4—and compare their performance with and without SELFI (Tab. 3). Across all configurations, SELFI consistently improves cross-dataset AUC: CLIP shows the largest gain with a 5.6% improvement, followed by ResNet34 with 2.6% and EfficientNet-B4 with 1.6%. These results demonstrate that SELFI is highly transferable and can be seamlessly plugged into diverse feature extractors, offering consistent improvements regardless of model capacity or architecture. This backbone-agnostic behavior underscores the practical utility of SELFI as a general-purpose deepfake detection enhancement module.

**Limitations.** Although SELFI adaptively fuses identity and visual features based on a learned relevance score, its effectiveness may be influenced by the following factors: (1) the quality of the extracted identity embeddings, which may be unreliable when the face is partially visible or captured from extreme angles (e.g., side profiles); and (2) when trained solely on a single manipulation type whose face identity cues are ineffective or non-transferable (e.g., NT and F2F), SELFI may suffer degraded cross-manipulation performance (see Sec. A.6 for details).

Furthermore, while our work focuses on identity-related bias as a primary factor, other sources of bias—such as background, ethnicity, or compression artifacts—have also been discussed in prior studies Yan et al. (2023a); Lin et al. (2024). A comprehensive analysis of these factors is beyond the scope of this work, but presents a valuable direction for future exploration.

## 6 CONCLUSION

This work presents SELFI, a novel framework for generalizable deepfake detection that explicitly and adaptively incorporates face identity information. Through extensive analysis, we show that identity features—when used indiscriminately—can act as both powerful cues and harmful biases, depending on the manipulation context. Motivated by this observation, SELFI separates identity cues from visual content and learns to fuse them based on per-sample relevance, leveraging their benefits while mitigating overfitting. Our method consistently outperforms strong baselines across both in-domain and cross-domain settings, and demonstrates robust performance across diverse backbones. Furthermore, ablation studies confirm that SELFI's gains stem from semantically meaningful identity information, rather than naive feature ensembling. We believe these insights offer a principled and practical direction for enhancing the generalization ability of deepfake detectors, and encourage future work to extend adaptive feature integration to other sources of bias beyond identity.

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

# SUPPLEMENTARY MATERIAL

## A  APPENDIX

### A.1  IMPLEMENTATION DETAILS

**Implementation Details for Tab. 1** All experiments follow the training configuration and preprocessing procedures defined by the DeepfakeBench Yan et al. (2023b) framework. We use CLIP as the backbone in our best-performing model. The input frames are resized to $224 \times 224$, and the model is trained with a batch size of 64 for 10 epochs. The hyperparameters $\alpha$ and $\beta$ are both set to 1. During training, we monitor the frame-level AUC on the validation set and save the model with the highest score. All experiments are conducted using five NVIDIA Titan RTX GPUs.

### A.2  DETAILS OF EXPERIMENTAL SETTING FOR SEC. 3

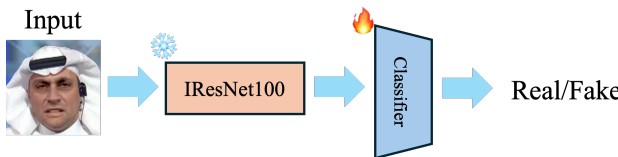

Figure 5: **Architecture of the identity-only detection setting.** A lightweight classifier is trained on top of frozen face identity embeddings extracted from a face recognition model, without any access to visual content. This setup is used to investigate whether face identity alone can support deepfake detection.

To investigate whether face identity alone can support deepfake detection, we design an identity-only detection setup where no visual content is used. Specifically, we extract identity embeddings from a pretrained face recognition model (IResNet100), which is frozen throughout training. A lightweight classifier head is then trained on top of these embeddings to classify whether the input is real or fake. The overall architecture is illustrated in Fig. 5. Since the output of IResNet100 is a 512-dimensional feature vector, we append a linear classifier that maps this 512-dimensional identity representation into a 2-class prediction space (real or fake).

We conduct cross-manipulation experiments using the FaceForensics++ dataset, focusing on four manipulation types: Deepfake (DF), FaceSwap (FS), Face2Face (F2F), and NeuralTextures (NT). In each setting, we train the classifier using identity features from one manipulation method and evaluate its generalization to the remaining ones. This setup allows us to examine how well identity-based signals transfer across manipulation types. The full results are reported in Tab. 5.

Table 5: **Cross-manipulation performance of identity-only detection.** Each model is trained using identity features extracted from a frozen face recognition model, without visual content, and evaluated across different manipulation types. Diagonal entries represent in-domain performance, while off-diagonal entries indicate cross-manipulation generalization. Results show that some manipulations (e.g., DF, FS) retain transferable identity cues, whereas others (e.g., NT) lead to poor generalization.

| Train | NT | FS | F2F | DF | Avg |
|-------|------|------|------|------|------|
| NT | **0.608** | 0.535 | 0.569 | 0.540 | 0.563 |
| FS | 0.509 | **0.846** | 0.565 | 0.745 | 0.666 |
| F2F | 0.524 | 0.509 | **0.826** | 0.523 | 0.595 |
| DF | 0.519 | 0.721 | 0.537 | **0.901** | **0.670** |
| All | 0.550 | 0.756 | 0.714 | 0.807 | 0.707 |

For all experiments, we use the standard CrossEntropy loss and train the classifier with the AdamW optimizer. The learning rate is set to 0.0002. Each model is trained for 10 epochs, and we save the best model based on frame-level AUC on the validation set.

## A.3 Video-level AUC Comparisoon

Table 6: **Video-level AUC Performance.** Scores for other methods are reported from Haliassos et al. (2022) and their respective original publications Zhao et al. (2023); Wang & Chow (2023); Zhang et al. (2024); Yan et al. (2024).

| Method | Publication | CDFv2 | DFDC | Avg |
|--------|-------------|-------|------|-----|
| LipForensics Haliassos et al. (2021) | CVPR'21 | 0.824 | 0.735 | 0.780 |
| FTCN Zheng et al. (2021) | ICCV'21 | 0.869 | 0.740 | 0.805 |
| RealForensics Haliassos et al. (2022) | CVPR'22 | 0.869 | 0.759 | 0.814 |
| ISTVT Zhao et al. (2023) | TIFS'23 | 0.841 | 0.742 | 0.792 |
| NoiseDF Wang & Chow (2023) | AAAI'23 | 0.759 | 0.639 | 0.699 |
| NACO Zhang et al. (2024) | ECCV'24 | 0.895 | 0.767 | 0.831 |
| LSDA Yan et al. (2024) | CVPR'24 | 0.911 | 0.770 | 0.841 |
| SELFI (Ours) | | 0.893 | **0.820** | **0.857** |

In addition to frame-level evaluations, we compare our method against recent state-of-the-art video-level detectors, as summarized in Tab. 6. Since our model operates at the frame level, we follow the evaluation protocol commonly used in other frame-level studies, where 32 evenly sampled frames are extracted from each video and individually classified. The final video-level prediction is then obtained by averaging the frame-level outputs. While our method slightly underperforms the strongest baseline on CDFv2, it achieves the best performance on DFDC and records the highest average video-level AUC overall. Notably, our method surpasses NACO Zhang et al. (2024)—the most recent and competitive video-level detector—by 2.6% in average AUC, despite relying solely on frame-level information without any explicit modeling of temporal dynamics. This demonstrates the strength of our spatial feature representation in generalizing across videos without temporal supervision.

## A.4 Details of Experimental Setting for Tab. 2

To analyze the contribution of each component within the SELFI framework, we perform a step-by-step ablation study under four configurations. The first setting disables all modules (FAIA, $\mathcal{L}_{\text{fs}}$, and IAFM), where the model is trained solely using the backbone feature. The second setting enables only FAIA, where identity embeddings are projected and simply concatenated with visual features before classification—without using the relevance predictor or soft fusion. In the third setting, both FAIA and IAFM are activated, allowing conditional fusion via the relevance predictor, but without the forgery-aware guidance loss. The final setting corresponds to the full SELFI model, which combines all modules.

In addition to CLIP, we also evaluate the ablations on two alternative backbones—EfficientNet-B4 and ResNet34—to assess the consistency of SELFI's modular benefits across architectures. The

Table 7: Ablation study of SELFI framework on EfficientNetb4 and ResNet34. We incrementally add FAIA, $\mathcal{L}_{\text{fs}}$, and IAFM to examine their contributions. Best results per backbone are **bolded**.

| Modules | | | EfficientNetb4 | | | | ResNet34 | | | |
|---|---|---|---|---|---|---|---|---|---|---|
| FAIA | $\mathcal{L}_{\text{fs}}$ | IAFM | CDFv2 | DFD | DFDCP | Avg | CDFv2 | DFD | DFDCP | Avg |
| ✗ | ✗ | ✗ | 0.747 | 0.823 | 0.685 | 0.751 | 0.739 | 0.810 | 0.690 | 0.746 |
| ✓ | ✗ | ✗ | 0.737 | **0.851** | 0.689 | 0.759 | 0.755 | 0.813 | 0.678 | 0.749 |
| ✓ | ✗ | ✓ | 0.721 | 0.818 | 0.695 | 0.745 | 0.739 | 0.814 | 0.730 | 0.761 |
| ✓ | ✓ | ✓ | **0.755** | 0.812 | **0.733** | **0.767** | **0.762** | **0.822** | **0.731** | **0.772** |

results are presented in Tab. 7, showing that each module incrementally contributes to performance improvements, and that the full SELFI configuration achieves the highest average AUC on both backbones. Notably, adding only FAIA yields a noticeable gain over the baseline, demonstrating that identity embeddings alone—when properly aligned—are useful for detection. Incorporating IAFM further improves performance by adaptively weighing identity relevance. Finally, the inclusion of the forgery-aware guidance loss $\mathcal{L}_{\text{fs}}$ consistently enhances results, confirming the importance of supervising identity transformation with task-specific signals. These trends are consistent across both EfficientNet-B4 and ResNet34, validating SELFI's backbone-agnostic effectiveness.

For both EfficientNet-B4 and ResNet34, we use input resolution of 256×256 and train models using the Adam optimizer with a learning rate of 0.0002, $\beta_1 = 0.9$, $\beta_2 = 0.999$, and weight decay of 0.0005. All models are trained for 10 epochs, and the best model is selected based on validation AUC. Both backbones are initialized with ImageNet pretraining.

## A.5 Details of Experimental Setting for Tab. 10

Table 8: Standalone performance of the Deepfake-Trained ResNet101 model used as an auxiliary source in Tab. 10. All results are reported as frame-level AUC.

| Method | FF++ | CDFv2 | DFD | DFDC | DFDCP | C-Avg. |
|---|---|---|---|---|---|---|
| ResNet101 | 0.977 | 0.720 | 0.781 | 0.693 | 0.680 | 0.719 |

All auxiliary sources used in Tab. 10 are extracted from the same backbone architecture, ResNet101, to ensure fair comparison. For the **Face Identity** extractor, we utilize a pretrained IResNet100 model, which applies Batch Normalization to its output features. To maintain consistency, we apply Batch Normalization to the output features of all other ResNet101-based extractors before feeding them into the IAFM module.

The **Deepfake-Trained** ResNet100 is trained separately using a standard deepfake classification setup. Its standalone performance (i.e., without fusion) is reported in Tab. 8 for reference. All auxiliary feature extractors are frozen during SELFI training and only their projected features are used for fusion with CLIP features.

## A.6 Ablation on Cross-Manipulation Comparison

SELFI is specifically designed for multi-manipulation real-world scenarios, where diverse generation techniques are encountered simultaneously. The core philosophy of SELFI is to explicitly disentangle the role of face identity in such environments, as stated in Hypothesis 2: DeepFake (DF) and FaceSwap (FS) exhibit transferable identity information, whereas NeuralTextures (NT) and Face2Face (F2F) contain method-specific or ineffective identity cues.

Given this design, SELFI inevitably faces challenges when trained on a single manipula-

Table 9: Cross-manipulation frame-level AUC results from Kashiani et al. (2025).

| Train | Method | DF | F2F | FS | NT |
|---|---|---|---|---|---|
| DF | GFF Luo et al. (2021) | 0.999 | 0.769 | 0.472 | 0.729 |
| | DCL Sun et al. (2022) | **1.000** | 0.771 | 0.610 | 0.750 |
| | IID Huang et al. (2023) | 0.995 | - | 0.638 | - |
| | SFDG Wang et al. (2023a) | 0.997 | 0.865 | 0.753 | 0.861 |
| | FreqDebias Kashiani et al. (2025) | 0.998 | **0.881** | 0.759 | **0.885** |
| | SELFI (Ours) | **1.000** | 0.630 | **0.808** | 0.583 |
| F2F | GFF Luo et al. (2021) | 0.892 | 0.991 | 0.613 | 0.648 |
| | DCL Sun et al. (2022) | 0.919 | 0.992 | 0.596 | 0.667 |
| | SFDG Wang et al. (2023a) | 0.974 | 0.994 | 0.735 | 0.726 |
| | FreqDebias Kashiani et al. (2025) | **0.984** | 0.994 | **0.744** | **0.765** |
| | SELFI (Ours) | 0.726 | **0.998** | 0.643 | 0.572 |
| FS | GFF Luo et al. (2021) | 0.702 | 0.687 | **0.999** | 0.499 |
| | DCL Sun et al. (2022) | 0.748 | 0.698 | **0.999** | 0.526 |
| | IID Huang et al. (2023) | 0.754 | - | 0.997 | - |
| | SFDG Wang et al. (2023a) | 0.817 | 0.773 | 0.995 | 0.609 |
| | FreqDebias Kashiani et al. (2025) | 0.838 | **0.789** | 0.998 | **0.635** |
| | SELFI (Ours) | **0.957** | 0.662 | 0.998 | 0.476 |
| NT | GFF Luo et al. (2021) | 0.885 | 0.498 | 0.743 | 0.988 |
| | DCL Sun et al. (2022) | 0.912 | 0.521 | 0.793 | 0.990 |
| | SFDG Wang et al. (2023a) | 0.917 | 0.709 | **0.836** | 0.997 |

tion and evaluated on unseen manipulations (cross-manipulation), as the model cannot fully leverage its adaptive identity-handling mechanism. Nevertheless, SELFI demonstrates two key strengths. First, it achieves state-of-the-art performance in most in-domain settings, with the exception of NT, which inherently lacks effective identity cues. Second, in cross-manipulation scenarios where identity information is transferable (e.g., between DF and FS), SELFI significantly outperforms prior methods. For example, when trained on DF and tested on FS, SELFI shows a 4.9% improvement over the previous state-of-the-art. Similarly, when trained on FS and tested on DF, it achieves an even larger 11.9% improvement.

In summary, SELFI excels at cross-manipulation detection when identity cues are transferable, while its performance is relatively weaker on manipulations like NT and F2F, where identity information is method-specific or ineffective. This observation further supports the importance of SELFI's adaptive framework in capturing identity semantics for robust and generalizable deepfake detection.

## A.7 ABLATION ON FUSION METHODS IN RELEVANCE PREDICTOR

The Relevance Predictor is a critical sub-module within the Identity-Aware Fusion Module (IAFM) of SELFI as provided in Sec. 4.2. Its primary role is to estimate the relevance score ($\rho$), which is then used as the weighting factor in the soft fusion process of IAFM. In our best-performing configuration, we adopt a concatenation (Concat) strategy, where the projected identity feature ($f_{\text{fi}}$) and the visual feature ($f_{\text{vis}}$) are concatenated and fed into the Relevance Predictor.

Below is a brief overview of the fusion strategies we explored. Except for *Dot*, all methods involve feeding the resulting feature into a shared MLP for relevance prediction:

(1) **Identity_mlp**: Uses only the projected identity embedding as input to the MLP.

$$z = f_{\text{fi}}$$

(2) **Visual_mlp**: Uses only the visual embedding from CLIP as input to the MLP.

$$z = f_{\text{vis}}$$

(3) **Dot**: Computes the scalar dot product between the two embeddings and applies a sigmoid, without an MLP.

$$\rho = \sigma \left( f_{\text{fi}}^{\top} f_{\text{vis}} \right)$$

(4) **Diff**: Uses the element-wise absolute difference between the two embeddings as input to the MLP.

$$z = |f_{\text{fi}} - f_{\text{vis}}|$$

(5) **Product**: Uses the element-wise product between the two embeddings as input to the MLP.

$$z = f_{\text{fi}} \odot f_{\text{vis}}$$

(6) **Add**: Uses the element-wise sum of the two embeddings as input to the MLP.

$$z = f_{\text{fi}} + f_{\text{vis}}$$

(7) **Concat**: Concatenates both embeddings and feeds the result into the MLP.

$$z = \text{Concat} \left( f_{\text{fi}}, f_{\text{vis}} \right)$$

As shown in Table 10, the *Concat* strategy achieves the best average performance (C-Avg = **0.846**), significantly outperforming other fusion methods. Single-source strategies such as *Identity_mlp* and *Visual_mlp* show lower performance on cross-manipulation datasets (e.g., CDFv2, DFDCP), indicating that both identity and visual information are necessary for robust generalization. The *Dot*

Table 10: **Comparison of Fusion Methods in Relevance Prediction.** The C-Avg column denotes the average performance across all datasets. The Concat strategy achieves the best overall performance.

| Fusion Method | FF++ | CDFv2 | DFD | DFDC | DFDCP | C-Avg. |
|---|---|---|---|---|---|---|
| Identity only | **0.981** | 0.810 | 0.833 | 0.770 | 0.806 | 0.805 |
| Visual only | 0.983 | 0.760 | 0.805 | 0.760 | 0.737 | 0.763 |
| Dot | 0.819 | 0.586 | 0.626 | 0.629 | 0.582 | 0.606 |
| Diff | 0.980 | 0.825 | 0.825 | 0.784 | 0.791 | 0.806 |
| Product | 0.980 | 0.763 | 0.868 | 0.755 | 0.784 | 0.793 |
| Add | 0.980 | 0.775 | 0.817 | 0.752 | 0.763 | 0.777 |
| Concat | 0.980 | **0.839** | **0.907** | **0.796** | **0.840** | **0.846** |

Table 11: **Effect of Projection Depth in FAIA.** The C-Avg column denotes the average performance across all datasets. The Single Projection strategy achieves the best overall performance.

| Method | FF++ | CDFv2 | DFD | DFDC | DFDCP | C-Avg. |
|---|---|---|---|---|---|---|
| 3-Layer (MLP-ReLU) | 0.975 | 0.759 | 0.848 | 0.763 | 0.790 | 0.790 |
| 2-Layer (MLP-ReLU) | 0.976 | **0.847** | 0.905 | 0.790 | 0.826 | 0.842 |
| 1-Layer (Single Projection) | **0.980** | 0.839 | **0.907** | **0.796** | **0.840** | **0.846** |

method, which lacks an MLP and directly applies a sigmoid, performs worst overall (C-Avg = 0.606), highlighting the importance of nonlinear interactions between features. Among element-wise operations, *Diff* and *Product* perform reasonably well but still fall short compared to *Concat*. These results confirm that explicitly combining the full representations of $f_{\text{fi}}$ and $f_{\text{vis}}$ through concatenation provides the most discriminative input for the Relevance Predictor.

## A.8 ABLATION ON IDENTITY EMBEDDING PROJECTION

In FAIA, a projection weight is utilized to transform existing face identity features to better suit the deepfake domain. We conducted an ablation study to compare three different projection methodologies, specifically investigating the effects of using a Multi-Layer Perceptron (MLP) for projection and observing phenomena as the layer depth increases.

Our findings, summarized in Tab. 11, indicate that the mere use of an MLP for projection results in a slight performance degradation. Furthermore, deeper layers lead to a more significant drop in performance. For instance, increasing the number of layers from two to three causes approximately a 5 percentage point decrease in performance based on the C-Avg metric.

This observation suggests that face identity features are inherently effective for deepfake detection. A shallower projection layer, which involves less information transformation, is therefore more beneficial. Conversely, a deeper layer leads to excessive transformation of the identity information, resulting in performance loss. Consequently, our research adopts the Single Projection method, which demonstrates superior performance.

