# OpenReview forum: "SELFI: Selective Fusion of Identity for Generalizable Deepfake Detection"
_ICLR.cc/2026/Conference — ICLR 2026 Conference Withdrawn Submission_

### Official Review · Reviewer_fkdR · 2025-10-22

**Soundness:** 3
**Presentation:** 2
**Contribution:** 3
**Rating:** 6
**Confidence:** 5

**Summary:**

This paper proposes to revisit the usage of identity information in deepfake detection, where two conflicting assumptions are widely adopted in the research community. It gives a brief demonstration on the FF++ dataset of the advantages and necessity of using identity in a decoupled manner. Then, this paper designs SELFI to use the helpful identity and get rid of the harmful identity. The experimental results show certain improvement on the existing datasets.

**Strengths:**

1.	The motivation of this paper is valuable and significant. There indeed exist two major perspectives in the current research community—one emphasizing the usefulness of identity information and the other questioning its necessity. Therefore, investigating this issue in depth and proposing further solutions is a meaningful and timely contribution.
2.	The proposed FAIA and IAFM modules are well-designed and demonstrate thoughtful architectural considerations.
3.	The paper makes an effort to provide theoretical explanations to support its claims, which enhances the overall rigor of the work.

**Weaknesses:**

1.	Limited validation on outdated deepfake methods.
The validation experiments should be conducted on more advanced deepfake generation techniques. This point is crucial, as the current experiments rely solely on the FaceForensics++ (FF++) dataset, which contains relatively outdated and simplistic methods from around 2019. The identity consistency in those earlier face-swapping approaches is inherently limited. This introduces a significant logical gap in the validation: it is possible that the observed effect arises because the fake identities in FF++ are not sufficiently realistic, leaving a perceptible distance for models such as IResNet.
In contrast, modern deepfake approaches—especially diffusion-based full-face generation models—can produce highly realistic and identity-preserving results. Under such conditions, reassessing whether “harmful or helpful identities” still exist becomes necessary for the validity of Hypothesis 1.
2.	Overly narrow scope of analysis (Lines 81–86).
Based on the concern above, the analysis should provide more general insights instead of being confined to the four outdated methods in FF++. Specifically, the paper should consider discussing broader manipulation categories—such as Face Swapping (FS), Face Editing (FE), and Entire Face Synthesis (EFS)—and analyze their potential associations with identity-related representations. See [3] for relevant definitions and conceptual inspiration.
3.	Insufficient evaluation datasets.
The experimental section should incorporate additional, more contemporary datasets such as DF40 [3], DiffusionFace [4], or WDF, to improve representativeness and strengthen the empirical conclusions.
4.	Incomplete literature review on face provenance and de-biasing methods.
The related work section currently lacks coverage of key studies on face retracing based on IID approaches (e.g., [5],[6]), as well as recent advances in bias mitigation in face recognition (e.g., [1],[2]). Including these would help position the work more accurately within the existing literature.
5. Line88-91, repeated sentences. The authors should carefully revise their paper for an improved presentation.

[1] Exploring Unbiased Deepfake Detection via Token-Level Shuffling and Mixing // AAAI’25

[2] ED4: Explicit Data-level Debiasing for Deepfake Detection // TIP’25

[3] Toward next-generation deepfake detection[J]. //NIPS24

[4] Diffusionface: Towards a comprehensive dataset for diffusion-based face forgery analysis

[5] IDRetracor: Towards Visual Forensics Against Malicious Face Swapping

[6] Deepreversion: reversely inferring the original face from the deepfake face

**Questions:**

1. In Hyp2, could you explicitly explain the so-called "way"?

---

### Official Review · Reviewer_zC2q · 2025-10-29

**Soundness:** 3
**Presentation:** 3
**Contribution:** 2
**Rating:** 4
**Confidence:** 5

**Summary:**

This manuscript first investigates the impact of facial identity cues in deepfake detection and then introduces the SELFI framework. Specifically, the Forgery-Aware Identity Adapter (FAIA) projects facial identity embeddings into a forgery-discriminative space, after which the Identity-Aware Fusion Module (IAFM) adaptively fuses the projected identity features with visual features based on a learned relevance score.

**Strengths:**

- This work highlights an interesting yet often-overlooked question in deepfake detection — how facial identity cues influence detection performance across different manipulation types — and provides a systematic empirical analysis to investigate this effect.
- The manuscript is well-structured and easy to follow, and the authors release the core code for reproduction.

**Weaknesses:**

**1. Limited conceptual novelty.**
While this work identifies an interesting research question regarding the impact of facial identity cues in deepfake detection, the overall novelty of the solutions to this question is moderate.
The idea of integrating facial identity embeddings from a frozen face recognizer into a detection model is conceptually similar to prior works, such as RepDFD[1].

**2. Performances not leading among SOTA.**

_a)_ Although SELFI achieves consistent improvements across several backbones in Table 3, its performance does not surpass the existing works under the same backbone configurations. For instance, SELFI w/ EfficientB4（Table 3）performs below LSDA on several datasets, which also adopts the EfficientB4 in its original implementation.

_b)_ Incomplete comparison with latest CLIP-based detectors. Since SELFI utilizes CLIP as one of its main backbones, it should include direct comparisons with state-of-the-art CLIP-based deepfake detectors, such as RepDFD[1], Forensics Adapter[2] and Effort[3]. The absence of these comparisons weakens the completeness of the evaluation.



[1] Standing on the Shoulders of Giants: Reprogramming Visual-Language Model for General Deepfake Detection. AAAI 2025.

[2] Forensics Adapter: Adapting CLIP for Generalizable Face Forgery Detection. CVPR 2025.

[3] Orthogonal Subspace Decomposition for Generalizable AI-Generated Image Detection. ICML 2025.

**Questions:**

According to the results in Table 2, the FAIA module contributes the most significant improvement to the SELFI framework.
Is this improvement consistent when using different face recognition models?
How would the performance change if BlendFace [4], a face recognition model specifically designed for the face-swapping task, were employed in FAIA?

[4] BlendFace: Re-designing Identity Encoders for Face-Swapping. ICCV 2023.

---

### Official Review · Reviewer_QJr6 · 2025-10-31

**Soundness:** 3
**Presentation:** 3
**Contribution:** 3
**Rating:** 2
**Confidence:** 5

**Summary:**

The paper proposes a method called SELFI, which integrates information from face recognition models for deepfake detection. The authors argue that identity information of faces is effective only for certain types of forgery methods. Therefore, they introduce a Forgery-Aware Identity Adapter (FAIA) to model identity information in deepfake detection, and an Identity-Aware Fusion Module (IAFM) that dynamically fuses identity and image information to adaptively use both during detection.
However, although the motivation of the proposed method is novel, it has not been sufficiently verified. In addition, the paper lacks a comprehensive comparison with other similar methods. Finally, I believe the authors should explore their motivation in more depth and provide more insight. Based on the above points, I think the paper shows some innovation, but it still needs further improvement. The current version is not ready for acceptance.

**Strengths:**

1. The proposed method is clearly presented with understandable figures and explanations.

2. The motivation is relatively novel. Although facial identity information has been used in previous works, its different effects on various forgery types are still not well explored. This paper attempts to study this issue.

3. The method is simple and direct. The design is reasonable under the given motivation and easy to follow.

**Weaknesses:**

1. Lack of analysis and comparison with similar methods, e.g., those that use implicit identity information [1] or explicit usage of identity [2].

2. Inappropriate experimental setup. In Table 4, the comparison among different auxiliary features does not show a clear advantage of the proposed method. Moreover, the authors did not conduct ablation studies specifically on the use of face identity features. For example, they could study how different face recognition models affect SELFI or test different fusion strategies.

3. Insufficient validation of the motivation.

    - In Figure 2(a), showing performance only on training data cannot prove that face features generalize across domains.

    - In Figure 2(b), the authors only verify on the old FF++ dataset with limited forgery types. It would be more convincing to include richer forgery types, such as those in DF40 [3], which includes FS, FR, and EFS types. Adding experiments on some of them would strengthen the motivation.


[1] Huang B, Wang Z, Yang J, et al. Implicit identity driven deepfake face swapping detection[C]//Proceedings of the IEEE/CVF conference on computer vision and pattern recognition. 2023: 4490-4499.

[2] Lin K, Lin Y, Li W, et al. Standing on the shoulders of giants: Reprogramming visual-language model for general deepfake detection[C]//Proceedings of the AAAI Conference on Artificial Intelligence. 2025, 39(5): 5262-5270.

[3] Yan Z, Yao T, Chen S, et al. Df40: Toward next-generation deepfake detection[J]. Advances in Neural Information Processing Systems, 2024, 37: 29387-29434.

**Questions:**

1. Can the authors show whether the $\rho$ values output by IAFM after training align with their motivational assumption? This would improve the paper.

2. Reference [2] also combines a face model with the CLIP model. It should be added to the references.

3. The number of benchmarks tested is limited. The authors could include results on more recent benchmarks.

4. Since facial features are high-level representations, they should be more robust. Does using or not using facial features affect the robustness of the model?

---

### Official Review · Reviewer_9qWn · 2025-11-01

**Soundness:** 2
**Presentation:** 2
**Contribution:** 2
**Rating:** 2
**Confidence:** 5

**Summary:**

This paper addresses the challenge of generalization in deepfake detection by analyzing the role of face identity cues. Authors propose Selective Fusion of Identity method to learn identity features for DFDC.

**Strengths:**

The SELFI design (FAIA + IAFM) is simple yet effective, obtaining improvement with multiple backbones.
Extensive experiments across FF++, Celeb-DF v2, DFD, DFDC, and DFDCP show consistent cross-dataset gains

**Weaknesses:**

Will the identity information limit the model's generalization ability, especially for open-world tasks involving individuals with random identities? Additionally, could the identity feature be affected by age or makeup?

The description of Figure 1 is confusing. What characteristics do the up and down arrows in different colors represent? How are they related to identifying helpful or harmful identity information?

As shown in Table 1, the improvements brought by the proposed method are relatively limited.

Additionally, the reported accuracy of the current method seems questionable, as many existing approaches[1] have already surpassed 99% on tasks such as FF++.

[1] Qiao T, Xie S, Chen Y, et al. Fully unsupervised deepfake video detection via enhanced contrastive learning[J]. IEEE Transactions on Pattern Analysis and Machine Intelligence, 2024, 46(7): 4654-4668.

**Questions:**

please refer to weakness

---

### Note · Authors · 2025-11-13

I have read and agree with the venue's withdrawal policy on behalf of myself and my co-authors.